# Costs associated with retinopathy of prematurity: a systematic review and meta-analysis

Hanna Gyllensten ,[1,2] Jhangir Humayun ,[1,2] Ulrika Sjöbom,[1,3] Ann Hellström ,[3] Chatarina Löfqvist[1,2,3]

¹Institute of Health and Care Sciences, University of Gothenburg, Goteborg, Sweden
²Centre for Person-Centred Care, University of Gothenburg, Goteborg, Sweden
³Department of Clinical Neuroscience, Institute of Neuroscience and Physiology, University of Gothenburg, Goteborg, Sweden

**Correspondence to**
Professor Hanna Gyllensten;
hanna.gyllensten@gu.se

## ABSTRACT

**Objectives** To review and analyse evidence regarding costs for retinopathy of prematurity (ROP) screening, lifetime costs and resource use among infants born preterm who develop ROP, and how these costs have developed over time in different regions.

**Design** Systematic review and meta-analysis

**Data sources** PubMed and Scopus from inception to 23 June 2021.

**Eligibility criteria for selecting studies** Included studies presented costs for ROP screening and the lifetime costs (including laser treatment and follow-up costs) and resource use among people who develop ROP. Studies not reporting on cost calculation methods or ROP-specific costs were excluded.

**Data extraction and synthesis** Two independent reviewers screened for inclusion and extracted data, including items from a published checklist for quality assessment used for bias assessment, summary and random-effects meta-analysis for treatment costs. Included studies were further searched to identify eligible references and citations.

**Results** In total, 15 studies reported ROP screening costs, and 13 reported lifetime costs (either treatment and/or follow-up costs) for infants with ROP. The range for screening costs (10 studies) was US$5–US$253 per visit, or US$324–US$1072 per screened child (5 studies). Costs for treatment (11 studies) ranged from US$38 to US$6500 per child. Four studies reported healthcare follow-up costs (lifetime costs ranging from US$64 to US$2420, and 10-year costs of US$1695, respectively), and of these, three also reported lifetime costs for blindness (range US$26 686–US$224 295) using secondary cost data. Included papers largely followed the quality assessment checklist items, thus indicating a low risk of bias.

**Conclusion** The costs of screening for and treating ROP are small compared with the societal costs of resulting blindness. However, little evidence is available for predicting the effects of changes in patient population, screening schedule or ROP treatments.

**PROSPERO registration number** CRD42020208213.

## STRENGTHS AND LIMITATIONS OF THIS STUDY

⇒ PubMed and Scopus were searched systematically.
⇒ Since manual search of reference lists and citations of the identified papers did not identify additional studies, the database search had good coverage of the topic of investigation.
⇒ The main limitations of this work were the exclusion of grey literature and the lack of analyses of publication bias for the meta-analysis.
⇒ Where lack of variance information in included studies hindered meta-analysis, guidance for synthesis in systematic reviews without meta-analyses were followed.

such as retinopathy of prematurity (ROP). ROP is characterised by abnormal neurovascular development and, in its worst forms, retinal detachment and blindness.[2] Although preventable, ROP is the leading cause of blindness in children worldwide,[3] a ranking associated with the survival of infants with extremely low gestational age and birth weight in some parts of the world, and use of unmonitored treatments with 100% oxygen in other regions.[2]

ROP management and treatment economics are still challenging in many health systems because of screening-associated costs, patient-related costs and medicolegal liability.[4] Thus, there is an urgent need for more concerted efforts to guide healthcare providers in how to use resources efficiently, both in developing economies during a phase of improving survival of preterm infants, such as in many parts of Africa,[5] and in countries like Sweden with major neonatal morbidities still affecting a large proportion of those who survive.[6]

Here, we present an overview of costs associated with ROP screening and treatment, examining the evidence related to costs for ROP screening and lifetime costs (including laser treatment and follow-up costs) and resource use among infants born preterm who

## INTRODUCTION

Improvements in neonatal care have resulted in increased survival among children born preterm,[1] but these infants are at risk of developing preterm-related morbidities

develop ROP. We also examine the trajectories of these costs over time in different regions in a meta-analysis.

## METHODS

This work followed the Preferred Reporting Items for Systematic Review and Meta-Analysis (ie, PRISMA),[7] with protocol available in PROSPERO (reference CRD42020208213).[8]

### Article search

PubMed and Scopus were searched (online supplemental eTable 1, 23 June 2021) to identify original research on costs for ROP, including full cost or cost increases associated with ROP, without restricting language, publication date or country. Papers were thus included if presenting costs for ROP screening or lifetime costs (including laser treatment and follow-up costs) and resource use among people who develop ROP. Lifetime costs can for example include follow-up healthcare costs but also productivity loss due to blindness or other cost components occurring due to visual impairment later in life. Articles that did not describe the cost calculation method were excluded, as were those not presenting the costs for the group with ROP separately.

Rayyan QCRI was used for handling duplicates and the selection of studies for inclusion. Two independent reviewers (JH and CL or HG) searched the databases, screened articles for eligibility, extracted data using a prespecified data extraction sheet (online supplemental eTable 2), and handsearched included studies (7 July 2021) to identify eligible references and citations. Conflicting views were resolved by discussion with a third reviewer (CL or HG).

The data extraction sheet included items (online supplemental eTable 2) from a published checklist for quality assessment of economic evaluations[9] including a core set of items relevant in assessing the risk of bias in included studies. The 19 checklist items covers design and methods, population and generalisability, as well as ethics and funding, answered as yes or no during the assessment. To aid reading, summary scores indicating the items answered as Yes for each paper were calculated, thus a high summary score indicates that many of the items were covered. Quality of evidence was rated on a scale from 1 to 5 for individual articles, according to: 1=for example, properly powered randomised controlled trials; 2=for example, prospective cohort studies; 3=for example, retrospective cohort studies; 4=case series with or without intervention or cross-sectional study; 5=for example, opinion of respected authorities.[10]

### Analysis

Conventional screening (excluding telemedicine costs), laser treatment and long-term follow-up costs were reported, respectively, accounting for ROP severity and differences over time and between countries. Identified costs were adjusted to 2020 US dollars (US$) using annual exchange rates[11] and the Organisation for Economic Co-operation and Development inflation factor.[12] After imputation of missing variance based on the percentage variance in studies presenting such information, treatment costs were summarised in a forest plot, by year and subgroups using the World Bank country classification based on gross national income per capita,[13] as cost levels can be expected to differ.

### Patient and public involvement

This project did not include patient or public involvement in developing the research questions, design, conduct, choice of outcome measures or recruitment.

## RESULTS

Of the 503 studies screened after duplicates from the databases were removed, 123 were assessed for eligibility based on full text, and 19 studies were included in the synthesis of results (online supplemental eFigure 1). Reasons for exclusion were absence of data on costs associated with ROP, lack of original data or inclusion of data related only to insurance payments or litigation. No additional studies were identified by a hand search of references and a Scopus search of citations of included studies. An overview of all included studies[14–32] is presented in table 1, including references to secondary cost sources.[33–39] In total, 15 studies covered screening costs and 13 reported lifetime costs (treatment and/or follow-up costs) for infants who developed ROP.

Twelve studies were conducted in high-income economies: seven in the USA, two in Canada and one each in the UK, Netherlands and France. Three studies were conducted in upper-middle-income economies: one each in Peru, Thailand and Brazil. Three studies were conducted in lower-middle-income economies: two in India and one in Iran. One study was conducted in both the United States and Mexico (table 1). All studies reported the economic analyses using either US dollars, euros or local currency. The patient populations in all studies were infants at risk for ROP, although the studies used different inclusion criteria based on gestational age at birth and birth weight. In addition, the ROP definition for stages and treatment criteria varied with the timing of the study and international guidelines for classification at that time.

### Risk of bias in included studies

The quality assessment indicated a high overall quality of the included studies (online supplemental eTable 3), with 16 of 19 of them fulfilling at least 16 of the assessed criteria. However, eight studies did not fulfil the criteria for discounting future costs and outcomes or for subjecting results to sensitivity analyses to address the effects of assumptions. In addition, 14 studies met criteria regarding the reporting of incremental analysis and potential conflicts of interest. Thus, overall, the assessment suggested a low risk of bias in the included papers,

**Table 1** Overview of studies included in this review

| # | First author (year) | Country (study period) setting | Study design | ROP definition | Sample size (% of infants with ROP treated) | Inclusion criteria | Mean cost per child with ROP (value year and currency as reported in the original publication) | Cost perspective: cost inclusion |
|---|---|---|---|---|---|---|---|---|
| 1 | Mohammadi (2021)[14] | Iran (2017) Data from Farabi eye hospital | Decision Analytical Model from case series | Threshold ROP | Total: 126 ROP: 126 | Randomly selected infants with treatment requiring ROP | Treatment: US$1107/infant | Unclear perspective: out-of-pocket charges* |
| 2 | Moitry (2018)[15] | France (2012 and 2014–2015) Data from two hospitals CHSF and Port-Royal | Retrospective, before-and-after study | Type 1 ROP | Not specified | GA <33 weeks or BW <1500 g | Screening: €37/exam | Health system: direct costs |
| 3 | Isaac (2018)[16] | Canada (2009–2014) Data from Ontario Ministry of Health and Long-Term Care | Retrospective cohort study (chart review) | Type 1 ROP | Total: 174 ROP: 64 Treated: 3 (5.6%) | BW<1500 g or GA<30 weeks | Screening HSN: C$346/exam (SD: C$306) Screening RVH: C$375/exam (SD: C$300) | Health system: direct costs (excluding equipment and maintenance) |
| 4 | Kelkar (2017a)[17] | India (2009–2011) Mobile ROP screening unit | Public health intervention† from case series | ICROP guidelines | Total: 104 ROP: 34 Treated: 5 (15%) | BW<1700 g or GA<34 weeks | Screening: US$240/exam‡ Identifying an infant with ROP: US$735/infant‡ Treatment: US$6500/infant | Health system: direct healthcare costs (including salaries and equipment) |
| 5 | Kelkar (2017b)[18] | India (2013–2015) Data from 5 NICUs | Public health intervention† from case series | ICROP guidelines | Total: 102 ROP: 32 Treated: 4 (15%) | BW <1700 g or GA <34 weeks | Screening: US$199/infant§ Identifying an infant with ROP: US$596/infant§ Treatment: US$4137/infant | Health system: direct costs (including salaries and equipment) |
| 6 | Rothschild (2016)[19] | Mexico and US (2014) Data from paediatric eye clinics and schools for the blind in Atlanta, Georgia and Mexico City Blindness costs from the literature[33] and other secondary sources. | Decision Analytical Model from case series | ROP caused blindness (WHO) | Total: 95 | BW <1500 g | US screening: US$981/infant Mexico screening: US$333/infant US treatment: US$4037/infant Mexico treatment: US$505/infant US follow-up: US$1538/infant Mexico follow-up: US$2214/infant US blindness cost: US$84586/infant Mexico blindness cost: US$24413/infant | Third party payer: charges (including labour and equipment) Societal costs: expenses for raising a blind child |
| 7 | van der Akker-van Merle (2015)[20] | Netherlands (2009) Data from NEDROP study and PRN database | Retrospective cohort study | ICROP guidelines | Total: 1380 ROP: 29 Treated: 17 (59%) | GA<32 weeks or BW <1500 g | Screening: €109/exam Treatment: €2755/infant | Health system: direct costs |

Continued

**Table 1** Continued

| # | First author (year) | Country (study period) setting | Study design | ROP definition | Sample size (% of infants with ROP treated) | Inclusion criteria | Mean cost per child with ROP (value year and currency as reported in the original publication) | Cost perspective: cost inclusion |
|---|---|---|---|---|---|---|---|---|
| 8 | Wongwai (2015) [21] | Thailand (2013) Hypothetical data and cohort Blindness costs using secondary data on annual government subsidies and utilities from the literature [34] | Decision Analytical Model from prospective cohort study | ET-ROP criteria | Total: 100 ROP: 9 | | Screening: THB 142/infant Treatment: THB (SE) 1053 (316)/infant Lifetime cost of blindness: THB 146,000 Telemedicine screening: THB 17,397/QALY (3% disc. rate) | Third party payer: charges (including labour and equipment) |
| 9 | Black (2015) [22] | US (2001–2010) Medical University of South Carolina | Retrospective cohort study | ROP stage 4 | Total: 4292 ROP: 77 Treated: 7 (100%) | GA: 23–37 weeks | Cost increase due to ROP if: GA (23 w): US$19 513 GA (mean, 34.3 w): US$23 121 GA (37 w): US$41 161 | Hospital: direct costs |
| 10 | Zin (2014) [23] | Brazil (2004–2006) 6 NICUs in Rio de Janeiro | Decision Analytical Model from case series and expert opinion | ICROP criteria | Total: 869 ROP: 70 Treated: 70 (100%) | BW <1500 g | Screening: US$18/infant Treatment: US$398/infant | Health system: direct costs (including labour and equipment) |
| 11 | Dave (2012) [24] | Peru (2009) Data from local hospital's NICU and from 2002 study [39] Secondary source for blindness costs [35] | Retrospective cohort study | ROP stage 1–5 with/ without plus disease | Total: 1239 ROP: 80 | | Screening and treatment: US$2496/infant Follow-up (three visits): US$54 ROP caused blindness: US$123,806/infant | Health system: direct costs (including equipment, facility, labour and supplies) Societal costs: expenses for blindness |
| 12 | Dunbar (2009) [25] | US (2004–2006) Medicare and Medicaid reimbursement data from California and Louisiana | Microsimulation model from retrospective cohort study | Type 1 ROP | Total: 515 ROP: 58 Treated: 58 (100%) | BW <1500 g or GA<28 weeks | Screening: US$93/exam Screening: US$316/infant Treatment w/o anaesthesia: US$1371/infant Screening and treatment: US$1565/QALY (3% disc. rate) | Third-party payer (Medicare and Medicaid): charges (excluding anaesthesia) |
| 13 | Kamholz (2009) [26] | US (2005) Data from ET-ROP study | Decision Analytical Model from randomised trial and expert opinion | Type 1 ROP | ROP: 357 | BW<1250 g or GA<32 weeks | Screening: US$189/exam (US$56–$251); treatment w/o anaesthesia: US$2423 (US$638–$3223) Anaesthesia: US$1849 (US$925–$3698) | Third-party payer: charges |

Continued

**Table 1** Continued

| # | First author (year) | Country (study period) setting | Study design | ROP definition | Sample size (% of infants with ROP treated) | Inclusion criteria | Mean cost per child with ROP (value year and currency as reported in the original publication) | Cost perspective: cost inclusion |
|---|---|---|---|---|---|---|---|---|
| 14 | Jackson (2008)[27] | US (2006) Data from CRYO-ROP and ET-ROP studies | Decision Analytical Model from randomised trial | Type 1 ROP | Refer to published data on 4099 infants (65.8% with ROP[36] and 6998 infants (68% with ROP[37] | BW <1251 g | Screening: US$160/exam Screening and treatment: US$4410/QALY (3% disc. rate.) | Third-party payer (Medicare): charges |
| 15 | Yanowitch (2006)[28] | US (2001–2004) Data from Dean A. McGee Eye Institute and OUHSC campus | Retrospective cohort study (chart review) | CRYO-ROP and ET-ROP criteria | Total: 259 ROP: 11 Treated: 1 (9%) | BW 1250–1800 g | Screening: US$230/infant Treatment: US$2000/infant | Third-party payer: charges |
| 16 | Castillo-Riquelme (2004)[29] | UK (1997–1998) Data from published data[38] and local NICU | Decision Analytical Model from case series and expert opinion | ROP stage 3 | ROP: 235 | GA<32 or BW<1501 g | Screening: £49/exam Screening: £279/infant Treatment: £540/one eye Treatment: £702/two eyes Follow-up (10 years): £786/infant | Health system: direct costs (including equipment and maintenance) |
| 17 | Lee (2001)[30] | Canada (1996–1997) Data from 14 NICU | Retrospective cohort study | Threshold ROP | Total: 16 424 | Different criteria at different NICU | Screening: C$236/infant Treatment: C$2655/infant | Health system: direct costs |
| 18 | Brown (1999)[31] | US (1998) Database from Pennsylvania | Microsimulation model from randomised trial | Threshold ROP | ROP: 291 Treated: 291 (100%) but only one treated eye per infant | BW<1251 g | Treatment: US$1452/infant Treatment consultation: US$140/exam Treatment: US$678/QALY (3% disc. rate) | Third-party payer: charges |
| 19 | Javitt (1993)[32] | US (1989) Medicare reimbursement data | Microsimulation model from retrospective cohort study | Threshold ROP or PNA 24 weeks from CRYO-ROP | Total: 18 220 ROP: 1000 Treated: 1000 (100%) | BW: 500–1249 g | Screening (first visit): US$84/exam Screening (subsequent visit): US$68/exam Screening (weekly): US$6045/QALY Screening (biweekly): US$3623/QALY Screening (monthly): US$2488/QALY | Third-party payer: charges (excluding equipment and personnel training cost) |

*Assumption based on methods description indicating cost data collected through survey to parents.
†Studies of the introduction of new screening programmes.
‡Screening costs and costs for identifying an infant with ROP are reduced by 22.6% to account for transport costs (ie, driver and cost of van and fuel to move equipment).
§Screening costs and costs for identifying an infant with ROP are reduced by 0.245% to account for transport costs (ie, fuel to move equipment).
BW, birth wt; GA, gestational age; HSN, Health Sciences North in Sudbury; NICU, neonatal intensive care unit; PNA, postnatal age; QALY, quality-adjusted life-years; ROP, retinopathy of prematurity; RVH, Royal Victoria Hospital in Barrie, Canada.

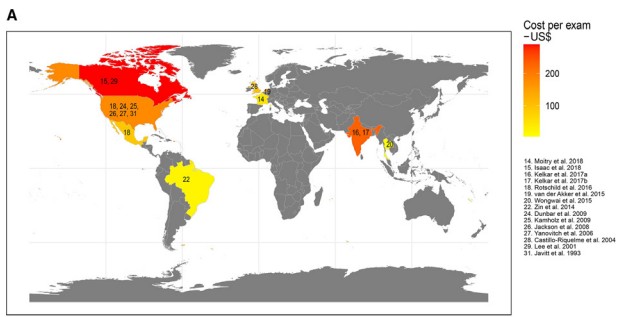

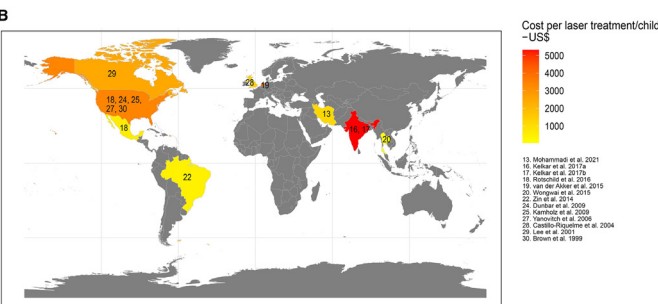

**Figure 1** Map of data availability and costs per (A) screening visit and (B) treatment. The map illustrates reported costs or means of reported costs per country for included studies in US$. In studies presenting only total screening cost per infant or by first/follow-up visits,[19 28 32] the cost level per screening was calculated under the assumption of four screening visits per infant. Where only screening cost per eye was reported,[30] it was duplicated to obtain the cost level per screening. In studies reporting only unit cost per treatment,[20 26] the unit cost was assumed to indicate the cost level of treatment per infant. where costs were reported separately for unilateral and bilateral treatment,[29] a weighted mean cost was calculated assuming that 75% of treatments were bilateral.

and also indicated where lack of reporting on potential conflicts of interest was most problematic. Quality of evidence ranged from 1 to 5 for individual articles, with articles most commonly based on data from retrospective cohort studies (evidence rating 3; nine publications).

## Costs for ROP screening

Studies reporting costs related to screening had different designs: six were retrospective cohort studies using medical chart review or register data,[15 16 20 24 28 30] nine developed economic models[19 21 23 25–27 29 31 32] and two were public intervention studies related to the introduction of ROP screening programmes.[17 18] Although the assessment indicated a low risk of bias, screening costs differed substantially among reporting countries (figure 1A).

Costs for routine ROP screening, excluding transportation costs, are reported in table 2. Ten studies reported a mean unit cost per screening of US$137 (range: 5–253). In addition, five studies reported a mean cost per screened child of US$553 (range: US$324–US$1072). Of these, two studies reported comparably low costs[21 23] for staff and equipment, whereas Rothschild et al[19] reported comparably higher costs in the US setting. One study also included transportation costs,[15] and when these costs

were removed, screening cost was comparably low. The other studies reported similar costs for screening per child (range: US$324–$602).[25 28 29]

Javitt et al[32] reported a mean unit cost of US$183 for a first screening and of US$149 for follow-up screening, whereas Lee et al[30] reported a mean unit cost of US$112 for screening one eye. Finally, two studies from India[17 18] reported screening costs of US$1003 and US$630, respectively, for identifying one child with ROP.

In studies comparing alternative screening or treatment options, no common comparator was identified. The incremental cost reported in Black et al[22] indicated a savings associated with higher gestational age at birth (table 1). Jackson et al[27] used economic modelling to estimate the cost-utility of ROP screening using telemedicine versus conventional ROP screening. Javitt et al[32] used modelling to compare weekly, biweekly or monthly screening.

## Costs for ROP treatment

In all, 14 studies reported costs related to the laser treatment of ROP (figure 1B). Four studies of treatment costs were retrospective cohort studies,[20 24 28 30] eight were modelling studies[14 19 21 23 25 26 29 31] and two were public intervention studies.[17 18] In addition, two of the included studies[31 32] reported costs for cryotherapy (not included in the analyses below).

Eleven studies reported total treatment costs per child, at a mean US$2442 (range: US$38–US$6500). Castillo-Riquelme et al[29] found unilateral treatment costs up to US$1165 and bilateral treatment costs up to US$1514, based partially on secondary data from Brown et al.[31] Two studies[20 26] cited unit costs of laser treatment of US$4065 and US$5661, respectively. Laser treatment costs are reported in table 2. Dave et al[24] described costs for screening and treatment combined (US$2962) in a cohort of children with blindness.

Accounting for the low assessed risk of bias but large expected variation based on cost-levels of individual countries, the meta-analysis by country classification (figures 2 and 3) estimated the average costs in high-income economies to US$2960 (95% CI US$2003 to US$3917). Corresponding figures were US$329 (95% CI US$9 to US$649) in upper-middle-income economies and US$3692 (95% CI US$670 to US$6715) in lower-middle–income economies, respectively. Most studies did not report variance of results, making publication bias analysis unfeasible. However, model diagnostics ($I^2$ and Cochrane Q) indicated high heterogeneity between studies within each country classification, which suggests that the results from the meta-analysis should be interpreted with caution.

## Follow-up costs and resource use among infants born preterm and developing ROP

Only four studies reported follow-up costs occurring after screening and treatment, and although the risk of bias was assessed as low, the reported results largely differed between studies. Castillo-Riquelme et al[29]

**Table 2** Costs for screening for ROP among preterm infants (in 2020 values)

| # | First author (year) | Screening costs | | Treatment costs | | |
| | | Mean per exam (US$) | Mean per infant (US$) | Mean per infant (US$) | Evidence rating | Cost inclusion |
|---|---|---|---|---|---|---|
| 1 | Mohammadi (2021)[14] | – | – | 1169 | 4 | Charges |
| 2 | Moitry (2018)[15] | 44 | – | – | 3 | Direct cost |
| 3 | Isaac (2018)[16] | HSN: 342 RVH: 371 | – | – | 3 | Direct cost not including equipment |
| 4 | Kelkar (2017a)[17] | 253 | – | 6500 | 4 | Direct cost including equipment and labour |
| 5 | Kelkar (2017b)[18] | 210 | – | 4137 | 4 | Direct cost including equipment and labour |
| 6 | Rothschild (2016)[19] | | US: 1072 Mexico: 362 | US: 4413 Mexico: 552 | 4 | Direct cost including equipment and labour |
| 7 | van der Akker-van Merle (2015)[20] | 160 | – | 4064* | 3 | Direct cost |
| 8 | Wongwai (2015)[21] | 5 | – | 38 | 2 | Charges including equipment and labour |
| 9 | Black (2015)[22] | – | – | – | 3 | – |
| 10 | Zin (2014)[23] | 20 | – | 450 | 5 | Direct cost including equipment and labour |
| 11 | Dave (2012)[24] | – | – | – | 3 | – |
| 12 | Dunbar (2009)[25] | 119 | 405 | 1759 | 3 | Charges |
| 13 | Kamholz (2009)[26] | 250 | – | 5661* | 5 | Charges |
| 14 | Jackson (2008)[27] | 205 | – | – | 1 | Charges |
| 15 | Yanowitch (2006)[28] | – | 324 | 2814 | 3 | Charges |
| 16 | Castillo-Riquelme (2004)[29] | 106 | 602 | Unilateral: 1165 Bilateral: 1514 | 5 | Direct cost including equipment and maintenance |
| 17 | Lee (2001)[30] | Unilateral: 112 | – | 2507 | 3 | Direct cost |
| 18 | Brown (1999)[31] | – | – | 2527 | 1 | Charges |
| 19 | Javitt (1993)[32] | First: 183 Follow-up: 149 | – | – | 3 | Charges |

Evidence rating indicates the quality of evidence rating of included studies: 1=for example, properly powered randomised controlled trials; 2=for example, prospective cohort studies; 3=for example, retrospective cohort studies; 4=case series with or without intervention or cross-sectional study; 5=for example, opinion of respected authorities.
*Unit cost per treatment.
HSN, Health Sciences North in Sudbury; ROP, etinopathy of prematurity; RVH, Royal Victoria Hospital in Barrie.

reported healthcare follow-up costs over 10 years of up to US$1695. Dave et al[24] reported a lifetime follow-up visit cost of US$64 and a blindness cost of US$146 952. Rothschild et al[19] reported lifetime follow-up healthcare costs of US$1681 (USA) and US$2420 (Mexico), whereas the costs for blindness were estimated to be US$92 460 (USA) and US$26 686 (Mexico). Wongwai et al[21] reported the lifetime costs of blindness to be US$224 295. In addition, Black et al[22] reported the costs per quality-adjusted life-year (QALY) associated with ROP and other comorbidities associated with being born preterm.

## DISCUSSION

The studies we identified could be grouped by whether they reported costs for screening, costs for treatment or costs (and QALYs) during long-term follow-up or even from a lifetime perspective. The cost range per ROP screening was US$5–US$253 per visit, or US$324–US$1072 per screened child. Costs for ROP treatment ranged from US$38–US$6500 per child. In addition, four studies reported healthcare follow-up costs, and three reported lifetime costs using secondary data on costs for blindness. Although quality assessment indicated a low risk of bias,

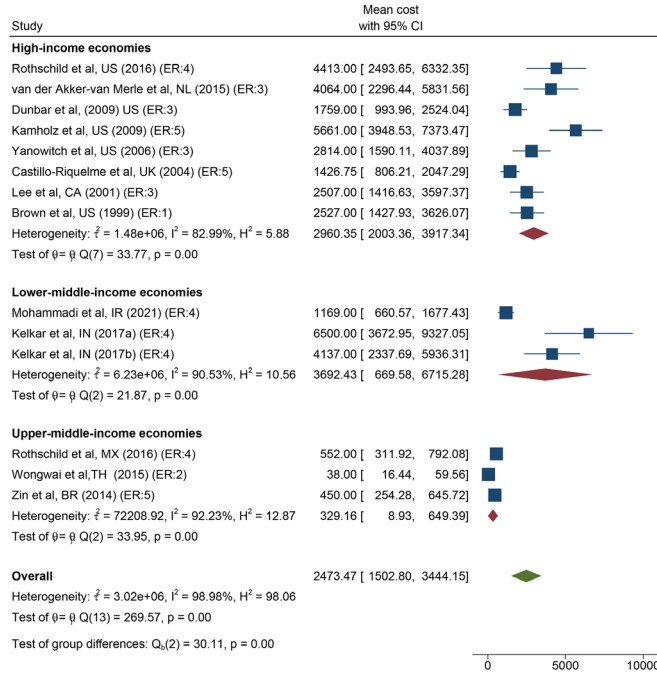

Figure 2 Forest plot of treatment costs, by country categorisation. In parentheses, ER of included studies: 1=for example, properly powered randomised controlled trials; 2=for example, prospective cohort studies; 3=for example, retrospective cohort studies; 4=case series with or without intervention or cross-sectional study; 5=for example, opinion of respected authorities. Country abbreviated according to ISO code. ER, evidence rating; REML, restricted maximum likelihood.

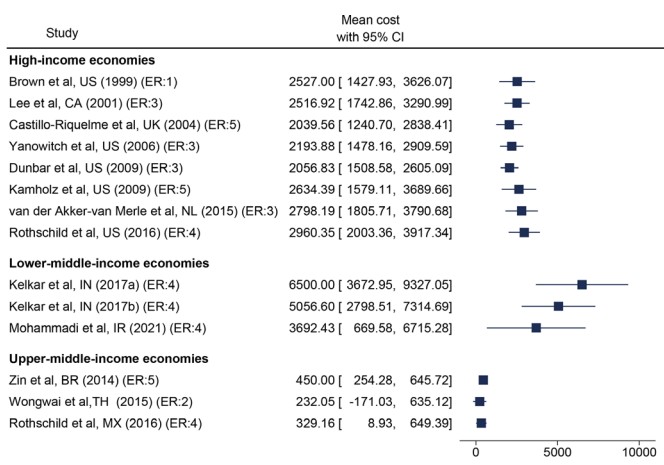

Figure 3 Forest plot of treatment costs, cumulative results by year and country categorisation. in parentheses, ER of included studies: 1=for example, properly powered randomised controlled trials; 2=for example, prospective cohort studies; 3=for example, retrospective cohort studies; 4=case series with or without intervention or cross-sectional study; 5=for example, opinion of respected authorities. Country abbreviated according to ISO code. ER, evidence rating; REML, restricted maximum likelihood.

comparisons between studies were challenging because of the lack of detailed cost and resource use data.

To our knowledge, this is the first systematic review of ROP costs. Included papers largely followed the quality assessment checklist items of a commonly used tool,[40] thus indicating a low risk of bias. However, few of the included articles reported disaggregated cost and resource use data or detailed the included cost components, as is recommended for economic evaluations.[41] The main limitations of this work were the exclusion of grey literature and the lack of analyses of publication bias for the meta-analysis. Guidance for reliability in systematic reviews of retinal disorder interventions[42] was fulfilled, but the standards for systematic reviews of costs and cost-effectiveness studies were not due to the lack of grey literature assessment.[43] Also, since costs were reported purely in a descriptive manner no sensitivity analyses were conducted for alternative categorisations of cost components or country classifications. While not a limitation specific to this analysis but rather of the lack of variance information in the included papers, the findings from the meta-analysis of treatment costs needs to be interpreted with caution after variance was imputed. This lack of variance information also made meta-analysis of screening costs unattainable, since no basis for imputation was available. Moreover, the search strategy and

databases are expected to cover largely English-language literature was limited to only two databases, but the reference and citation search yielded no additional studies to include. Thus, we expect our findings to represent a good overview of the available evidence, and that regardless the reservations associated with the meta-analysis to represent current knowledge about costs related to screening and treatment of ROP.

Cost components for ROP screening included staff salaries/time, equipment and maintenance, supplies and staff training. Screening costs for ROP were low compared with other associated costs and, with few exceptions, of the same order of magnitude in the included studies. Exceptions were probably attributable to salary differences.

Screening access and schedules vary between countries.[44] With the possible exception of Javitt et al,[32] the included studies provided little evidence for how casemix and alternative screening schedules affect costs for screening. Savings are expected, however, and a modelling study using published cost data calculated an annual cost savings from reduced screening of US$3 million in the USA.[45] However, with low screening costs, the main benefit is reduced discomfort for the infants and reduced travel costs, which can be substantial.[15] The most considerable potential for savings on screening is probably increasing gestational age. US data indicate that ROP frequency increased over time, particularly in infants born very preterm,[46] and infants of lower gestational age usually both require more screening visits and have more severe ROP.[47] Potential savings have been reported from screening using telemedicine (compared with transporting infants to a specialised hospital),[15] or

using bedside screening with mobile equipment instead of moving the infants to a specific screening facility[48]; however, this review did not consider these aspects.

Treatment costs were low compared with the costs for follow-up, with Brazil, Mexico and Peru having substantially lower treatment costs than the other countries. Both Javitt et al[32] and Brown et al[31] reported low costs for the historically used cryo treatment, at approximately 63% of that for laser treatment. For laser treatment, the cost range was US$2304–$6864 per treated child. None of the studies included the more recent antivascular endothelial growth factor (VEGF) therapy. Moreover, no study reported costs based on ROP stages, age of treated infants, or plus disease status.[49] Thus, studies provide little guidance on how treatment costs will develop over time as more infants of lower gestational age survive.

Variation among studies in whether one or two eyes were treated made comparisons less relevant, which may reflect the unilateral schedule used in the historically influential Cryo-ROP study.[50] However, Swedish registers indicate that bilateral treatment is common (76% of initial treatments and 97% overall)[47] and that retreatment is more frequent among infants with very low gestational age[51] and those treated exclusively with anti-VEGF.[47]

When examining ROP treatment, cost components included staff salaries/time, equipment and maintenance, supplies and staff training. Sometimes anaesthesia costs were reported separately or excluded. Transportation was also a considerable cost component in relation to treatment.[20] Other potential costs that were not measured include those for the added time spent in hospital or intensive care, including parental leave, during treatment. Many studies reported only total charges, which are expected to be higher than costs to the healthcare provider. However, use of charges as opposed to costs was not an obvious cause of variation here. Two studies from India[17 18] reported high costs compared with other studies of both costs and charges, possibly because of some transportation costs remaining as part of additional components. Thus, the apparent decrease in costs over time in the lower-middle-income economies seen in the meta-analysis should be interpreted with caution.

Although ROP results in high costs throughout life, this outcome is primarily based on secondary data for blindness. As the leading cause of preventable childhood blindness[52] and probably the leading cause of childhood blindness in middle-income countries,[53] ROP should be associated with much of the estimated costs of blindness. Moreover, it has been argued that costs for blindness do not differ by cause.[54] Little evidence was available on follow-up after successful, or partially successful, treatment of ROP. Dave et al[24] indicated three healthcare visits over the first 7 years of life, whereas Castillo-Riquelme et al[29] did not differentiate visits based on treatment or ROP stage. Rothschild et al included transportation costs, white canes, Braille equipment and supplies,[19] but disregarded other costs among children retaining sight. Thus, although costs differ by the severity of visual impairment,[55]

studies of ROP costs do not tend to report this more detailed level of sight. The current knowledge does not inform potential savings or inform subsidy decisions for ROP treatment developments that can save a little more sight. Taken together, the short follow-up underestimates the total impact of blindness,[56] and not accounting for visual impairment results in underestimating the financial impact of ROP.

There is a need for comprehensive knowledge about the costs of ROP, both during the introduction of new ROP screening programmes and in countries with established programmes that are now redistributing resources to handle the increasing survival of very preterm infants with high disease burden. In addition to relevant cost components of ROP (online supplemental eFigure 2), complementary studies of the benefits of various neonatal preventative strategies, including oxygen delivery, are warranted because evidence of the costs resulting from conditions such as bronchopulmonary dysplasia is also lacking.[57] Such studies should follow state-of-the-art methods for conduct and reporting of health economic studies.

## CONCLUSIONS

Although costs of screening and treating ROP are substantial for health systems, they are small compared with the follow-up costs to society of resulting blindness. However, little evidence is available to support predictions about the consequences of changes in the patient population, screening schedule or treatment regimens for ROP.

**Acknowledgements** Thanks to SF Edit, a professional scientific-editing service, for language editing.

**Contributors** All authors contributed to the design of the study. HG, JH and CL designed the database search and data extraction methods. JH and CL undertook the literature search, assessed studies for eligibility, and extracted data. In case of disagreement, assessments were made in discussion with HG. AH contributed clinical expertise on preterm infants and morbidity. HG, JH, US and CL discussed the data and interpreted the results. HG, JH and CL drafted the manuscript. All authors critically reviewed and approved the final manuscript. HG is the guarantor of the study and thus had full access to all the data in the study and takes responsibility for the integrity of the data and the accuracy of the data analysis.

**Funding** HG was financed by the Swedish Research Council (#2016-01131). JH was financed by the University of Gothenburg Centre for Person-Centred Care (GPCC) teaching assistant program. US was supported by The De Blindas Vänner. AH was supported by The Wallenberg Clinical Scholars, The Swedish Research Council (#2020-01092), the Gothenburg County Council (ALF project, #426531), the Gothenburg Medical Society, and De Blindas Vänner. CL was supported by The Wallenberg Clinical Scholars.

**Map disclaimer** The inclusion of any map (including the depiction of any boundaries therein), or of any geographic or locational reference, does not imply the expression of any opinion whatsoever on the part of BMJ concerning the legal status of any country, territory, jurisdiction or area or of its authorities. Any such expression remains solely that of the relevant source and is not endorsed by BMJ. Maps are provided without any warranty of any kind, either express or implied.

**Competing interests** HG is employed part-time by IQVIA, which is a contract research organisation that performs commissioned pharmacoepidemiological studies. Thus, its employees have been and currently are working in collaboration with several pharmaceutical companies. JH reports no competing interests. AH holds stock/stock options in Premalux AB and has received consulting fees from Takeda Inc. CL holds stocks in Premalux AB.

**Patient and public involvement**  Patients and/or the public were not involved in the design, or conduct, or reporting, or dissemination plans of this research.

**Patient consent for publication**  Not applicable.

**Provenance and peer review**  Not commissioned; externally peer reviewed.

**Data availability statement**  All data relevant to the study are included in the article or uploaded as online supplemental information. Original data are available in the reviewed publications, which are all cited. Additional data from the data extraction performed are available on reasonable request from the corresponding author, including author template data collection forms, data extracted from included studies, data used for all analyses, analytic code, and any other materials used in the review.

**Open access**  This is an open access article distributed in accordance with the Creative Commons Attribution 4.0 Unported (CC BY 4.0) license, which permits others to copy, redistribute, remix, transform and build upon this work for any purpose, provided the original work is properly cited, a link to the licence is given, and indication of whether changes were made. See: https://creativecommons.org/licenses/by/4.0/.

**ORCID iDs**
Hanna Gyllensten http://orcid.org/0000-0001-6890-5162
Jhangir Humayun http://orcid.org/0000-0002-0507-8216
Ann Hellström http://orcid.org/0000-0002-9259-1244

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
