## [Reviewer comments · BMJ Open]

ARTICLE DETAILS

TITLE (PROVISIONAL)	Costs Associated with Retinopathy of Prematurity: A Systematic Review and Meta-analysis
AUTHORS	Gyllensten, Hanna; Humayun, Jhangir; Sjöbom, Ulrika; Hellström, Ann; Löfqvist, Chatarina

VERSION 1 – REVIEW

REVIEWER	Gama, Rita Hospital da Luz
REVIEW RETURNED	29-Jan-2022

GENERAL COMMENTS	congratulations for this interesting study. Figures 1 and 2 give an important perspective of the results. page 5 of 53, line 14-15: "ROP is the leading cause of blindness worldwide", it should mention in children page 10 of 53, line 26-49: in this section, please mention from which country the author is from. page 37 of 53, figure 3: what was the criteria for the country categorization (high-income economy, upper-middle-income economy, etc)? It should be mentioned. page 37, figure 3: specifying the original country from each study
---

REVIEWER	Senjam, Suraj Singh All India Institute of Medical Sciences, Community Ophthalmology, Dr. Rajendra Prasad Centre for Ophthalmic Sciences
REVIEW RETURNED	16-Feb-2022

GENERAL COMMENTS	This article is a good peice of work. There are few comments on the manuscript. The author can consider those at the time of revision.
--

REVIEWER	Liu, Jin Vertex Pharmaceuticals Incorporated
REVIEW RETURNED	19-Jul-2022

GENERAL COMMENTS	Thanks for the opportunity to review the manuscript. This paper performs a systematic review and meta-analysis of the cost associated with Retinopathy of Prematurity (ROP) screening and treatment. First, one primary outcome of a meta-analysis is the forest plots presented in the online document. Therefore, these plots need to be moved to the main text with the necessary interpretation regarding the heterogeneity. In addition, the forest plots were only provided for treatment costs. It is also better to add forest plots to the cost of screening.
--

	Second, for the treatment costs that were cumulative results by year, the number of years followed by each study should be reported.
--	--

VERSION 1 – AUTHOR RESPONSE

Reviewer: 1

Dr. Rita Gama, Hospital da Luz

Comments to the Author:

congratulations for this interesting study. Figures 1 and 2 give an important perspective of the results.

Response: Thank you.

page 5 of 53, line 14-15: "ROP is the leading cause of blindness worldwide", it should mention in children page 10 of 53,

Response: This has been corrected according to suggestion.

line 26-49: in this section, please mention from which country the author is from.

Response: We have clarified which countries are indicated in these publications.

page 37 of 53, figure 3: what was the criteria for the country categorization (high-income economy, upper-middle-income economy, etc)? It should be mentioned.

Response: It has been clarified in the methods section that this is the World Bank classification, which is based on gross national income per capita.

page 37, figure 3: specifying the original country from each study

Response: ISO country code has been added in figures 2 and 3.

Reviewer: 2

Dr. Suraj Singh Senjam, All India Institute of Medical Sciences Comments to the Author:

This article is a good piece of work. There are few comments on the manuscript. The author can consider those at the time of revision.

Comments

The present manuscript is well written. It includes all essential components to construct a systemic review and meta-analysis. I appreciate all authors for this piece of good work; it is an important study when the prevention of ROP blindness is concerned. Some comments are given below that the author can consider while revising it.

Response: Thank you.

The author can define or clarify what is Lifetime costs in the study or mention them in the discussion. Such a cost may be not applicable in all children with ROP, though screening and treatment cost applies to the majority of them. For children with total blindness due to ROP (stage 5), lifetime cost also may include the loss of productivity or opportunity due to blindness, etc.

Response: We added a description of lifetime costs in the methods section, page 5.

Page 8, the last line, regarding the transportation cost, the author can clarify if possible whether this cost was incurred by providers or by the family of ROP kids. Often family members have to travel far away from the base hospital where screening and treatment facilities are available, whereas in some

cases, providers from tertiary level facilities visit lower-level health facilities having newborn care centers using a mobile screening van for ROP.

Response: We agree that this was unclear and have now removed this from the sentence indicated but clarified what was excluded in the caption to table 1. The excluded costs for transportation relate to costs for driving the equipment around, in evaluations of telemedicine interventions.

The author can explain briefly the second part of the Meta-analysis table (page 38) in the text, it seems indicating the change in cost over time in different economic regions. Can it be a representative cost for each region? There is a huge reduction in the cost associated with ROP in Low Middle Income Economies over time (page 38), perhaps the author can share possible reasons for this in the discussion.

Response: We believe this is at least partly caused by transportation costs not being fully excluded from the results presented for the two studies from India and agree that the cumulative cost in that category is difficult to interpret. We added a sentence about this in the discussion, page 12.

Will there be any effect on the ROP cost base on the years of studies being conducted? There might be a change in ROP related cost over the periods of time.

Response: As stated in the analysis section, we agree and have therefore kept the analyses to similar treatment regimens and reported all costs for a common value year after adjusting for different cost levels between countries. The remaining difference over time should be caused by differences between countries and over time, but also to a large extent on differences in the cost calculations made in the included studies.

Page 11, lines 38-40, the same being repeated on page 12, the last para. Please check it.

Response: These sentences are very similar but refer to screening and treatment, respectively. We have changed the sentence about cost components for treatment to improve reading.

In the discussion, the author can share with what are potential factors that lead to the high cost associated with ROP in LMICs compared to other countries, using existing evidence if any, and also potential savers. This is important because blindness due to ROP in LMICs increases significantly over the last few years.

Response: We agree and believe that it became clearer after the above addition about uncertainty in interpretation of the development of costs within the LMIC-category over time, as it comes out in the meta-analysis (page 12).

The author can briefly talk about sensitivity analysis in the manuscript, though publication bias analysis is not feasible.

Response: We added a sentence about the lack of sensitivity analyses in the discussion, page 10.

Reviewer: 3

Dr. Jin Liu, Vertex Pharmaceuticals Incorporated Comments to the Author:

Thanks for the opportunity to review the manuscript. This paper performs a systematic review and meta-analysis of the cost associated with Retinopathy of Prematurity (ROP) screening and treatment.

Response: Thank you.

First, one primary outcome of a meta-analysis is the forest plots presented in the online document. Therefore, these plots need to be moved to the main text with the necessary interpretation regarding the heterogeneity. In addition, the forest plots were only provided for treatment costs. It is also better to add forest plots to the cost of screening.

Response: The forest plots are presented in the main text (figures 2 and 3), but we agree that it's difficult to see that in the pdf. We expect that will be easier to see in a final publication.

We agree that it would be good to be able to also do a meta-analysis for the costs of screening, but this would only be possible at all for the cost per screening occasion, which in most instances has been treated as a unit cost in the studies. Only one study report variance data for costs per screening, thus it is difficult to know if this is representative for all studies. Thus, the forest plot does not add information compared to what is reported as cost range (“US\$137 (range: 5–253)”), page 8. Few studies report information about cost per screened child, which would otherwise have been interesting to do meta-analysis for, and none of those reported variance data.

Second, for the treatment costs that were cumulative results by year, the number of years followed by each study should be reported.

Response: Unfortunately, many of these studies include a range of years, or only report the year of data extraction from a database, making the figure very busy if including also time of data collection (as can be seen in table 1). Looking at the reported study periods in each paper, we can see that very few changes would be made to the order of papers in that figure if sorting instead on year of extracted data. For lower-middle-income economies and upper-middle-income economies the sorting would be identical, while for high-income economies some overlap occurs where one paper covers several years, and another has extracted data one of these years or in close proximity to. Brown (extracted 1998) and Lee (1996-1997) studies could thus change order, as well as Dunbar (2004-2006) and Kamholz (extracted 2005) studies. Otherwise, the sorting would be identical. We have added a sentence about this in the discussion, page 11.

VERSION 2 – REVIEW

REVIEWER	Senjam, Suraj Singh All India Institute of Medical Sciences, Community Ophthalmology, Dr. Rajendra Prasad Centre for Ophthalmic Sciences
REVIEW RETURNED	16-Sep-2022
GENERAL COMMENTS	The manuscript can be endorsed for publication
REVIEWER	Liu, Jin Vertex Pharmaceuticals Incorporated
REVIEW RETURNED	15-Sep-2022
GENERAL COMMENTS	I think the authors have already addressed the comments in the previous version.